# NeoMUST: an accurate and efficient multi-task learning model for neoantigen presentation

Wang Ma[1], Jiawei Zhang[1], Hui Yao[2]

Accurate identification of neoantigens is important for advancing cancer immunotherapies. This study introduces Neoantigen MUlti-taSk Tower (NeoMUST), a model employing multi-task learning to effectively capture task-specific information across related tasks. Our results show that NeoMUST rivals existing algorithms in predicting the presentation of neoantigens via MHC-I molecules, while demonstrating a significantly shorter training time for enhanced computational efficiency. The use of multi-task learning enables NeoMUST to leverage shared knowledge and task dependencies, leading to improved performance metrics and a significant reduction in the training time. NeoMUST, implemented in Python, is freely accessible at the GitHub repository. Our model will facilitate neoantigen prediction and empower the development of effective cancer immunotherapeutic approaches.

## Introduction

Neoantigens arising from the somatic mutations in cancer cells can elicit a tumor-specific immune response (Tn & Rd, 2015). They are promising targets for immunotherapies such as cancer vaccines (Sahin & Türeci, 2018), adoptive T-cell therapy (Tran et al, 2017), and immune checkpoint inhibitors (Sharma & Allison, 2015). The presentation of neoantigens through MHC molecules on the surface of antigen-presenting cells (APCs) is crucial in activating immune systems.

Accurate prediction of neoantigen presentation using genome sequencing and bioinformatics algorithms is critical for the successful development of cancer immunotherapies, and yet remains challenging. The complexity is rooted in a series of biological processes, including processing mutated proteins into short peptides, transporting them to the endoplasmic reticulum, loading them onto MHC molecules, and transporting them to the cell surface (Tn & Rd, 2015).

The state-of-the-art approaches for predicting neoantigen presentation through MHC class I (MHC-I) molecules use ensemble models, including NetMHCpan4.1 (Reynisson et al, 2020) and MHCflurry2.0 (O'Donnell et al, 2020). Both trained multiple artificial neural networks using binding affinity (BA) data and mass spectrometry (MS) eluted ligand (EL) data, and subsequently combined individual predictions using a weighted average. The ensemble models improved the accuracy of the prediction in large benchmark datasets, compared with other machine learning methods, such as the stabilized matrix method, the hidden Markov model, and the quantitative structure–affinity relationship–based regression model (Mei et al, 2020; Peters et al, 2020).

Recently, deep learning models have been proposed to predict neoantigen presentation. Some examples include DeepHLApan that applied the recurrent neural network–based method (Wu et al, 2019) and MixMHCpred2.2 that integrated convolutional neural networks and long short-term memory (LSTM) networks (Gfeller et al, 2023). They achieved performance comparable to that of ensemble models (Wu et al, 2019; O'Donnell et al, 2020).

Despite significant progress, there are two major issues with the existing models. The first issue is that the neoantigen–MHC binding measured by BA data and neoantigen presentation measured by MS data represent distinct prediction tasks involving relevant but different biological processes. The existing models training a single neural network architecture with a single loss function to unify these tasks are ineffective in capturing task-specific information. The second issue is that ensemble models increase the overall complexity, and thus require significantly more computational resources to train big datasets than individual models.

To address these issues, we developed our "Neoantigen MUlti-taSk Tower (NeoMUST)" model for the prediction of neoantigen presentation through MHC-I molecules. NeoMUST applied a multi-task learning (MTL) approach (Ruder, 2017 Preprint) and predicted the neoantigen presentation (NP) as its main task (Task A in Fig 1) and the neoantigen–MHC binding as an auxiliary task (Task B in Fig 1). Specifically, NeoMUST has three features that improve performance over the previous algorithms: (1) NeoMUST captures and leverages task-specific information across two related tasks and learns to identify commonalities and differences between them to improve performance; (2) it is optimized for individual loss functions that balance two tasks; and (3) it significantly reduces training time and thus improves its scalability for big datasets.

[1]Fresh Wind Biotechnologies Inc. (Tianjin), Tianjin, China    [2]Fresh Wind Biotechnologies USA Inc., Houston, TX, USA

Correspondence: hui.yao@freshwindbiotech.com

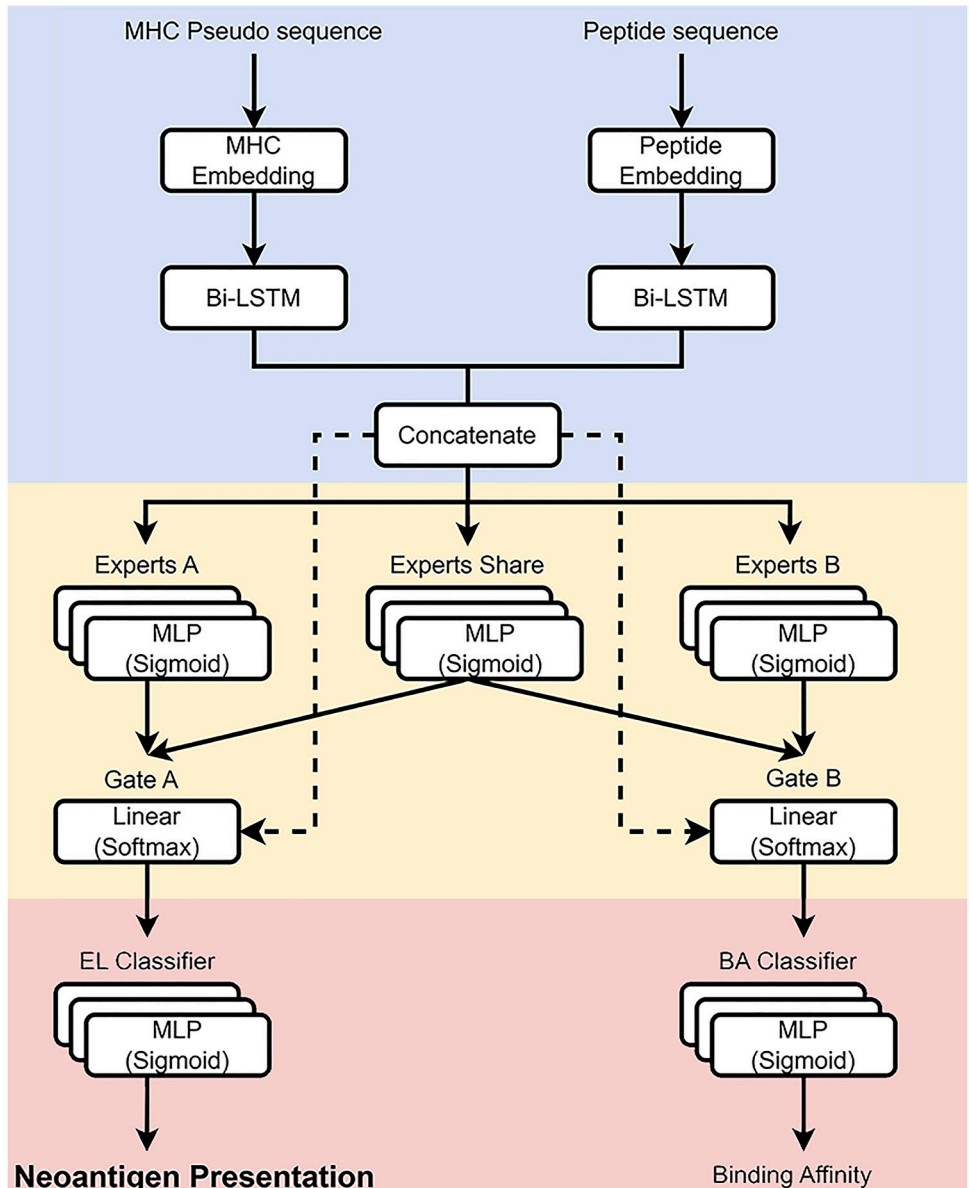

**Figure 1.  Schematic representation of the NeoMUST model architecture.**
The algorithm comprises three layers: the embedding layer (blue area), the expert layer (yellow area), and the prediction layer (red area). Peptide sequences and MHC-I pseudo-sequences are converted into high-dimensional vectors by the embedding layer, using bidirectional long short-term memory networks to extract sequence features. The merged feature matrices are then processed by three task-specific expert neural networks in the expert layer, with shared information integrated through gate networks. The prediction layer consists of two multi-layer perceptron networks for classifying neoantigen presentation and predicting binding affinity.

## Results

In this study, we developed an MTL model, NeoMUST, for accurate prediction of the neoantigen presentation in cancers. NeoMUST showed commendable performance in the benchmark studies. First, it exhibited comparable results to the state-of-the-art algorithms for the prediction of neoantigen presentation in a multi-allelic test set of MS data, TeSet-1 (O'Donnell et al, 2020). TeSet-1 represented a de facto clinical practice of up to six MHC-I alleles expressed in one individual, consisting of 76 individuals with a total of 9,158,100 peptide–MHC-I pairs with 1% of true positives. When focusing on the accuracy of positive prediction, the NeoMUST NP model outperformed both NetMHCpan4.0 EL and MHCflurry2.0 binding affinity (BA) models, exhibiting the larger area under the

precision–recall curve (AUC-PRs) with 40 samples versus 36 (Fig 2A) and 58 samples versus 18 (Fig 2B), respectively. A comparison with the MHCflurry2.0 presentation score (PS) model, which integrated antigen processing (AP) with the BA model, revealed a slightly inferior performance for NeoMUST NP, showing 31 samples compared with 45 (Fig 2C). The median (0.357) of AUC-PRs of NeoMUST NP was significantly higher than that of MHCflurry2.0 BA (median = 0.276, $P < 0.001$), and on par with NetMHCpan4.0 EL (median = 0.330, $P = 0.099$) and MHCflurry2.0 PS (median = 0.365, $P = 0.040$), shown in Fig 2D. Similarly, the NeoMUST NP model exhibited enhanced performance for top predictions, assessed by the positive prediction value (PPV 1%), representing the proportion of true positives within the top 1% of predicted scores per sample. Detailed results are presented in Fig S1A–D. Furthermore, to measure overall

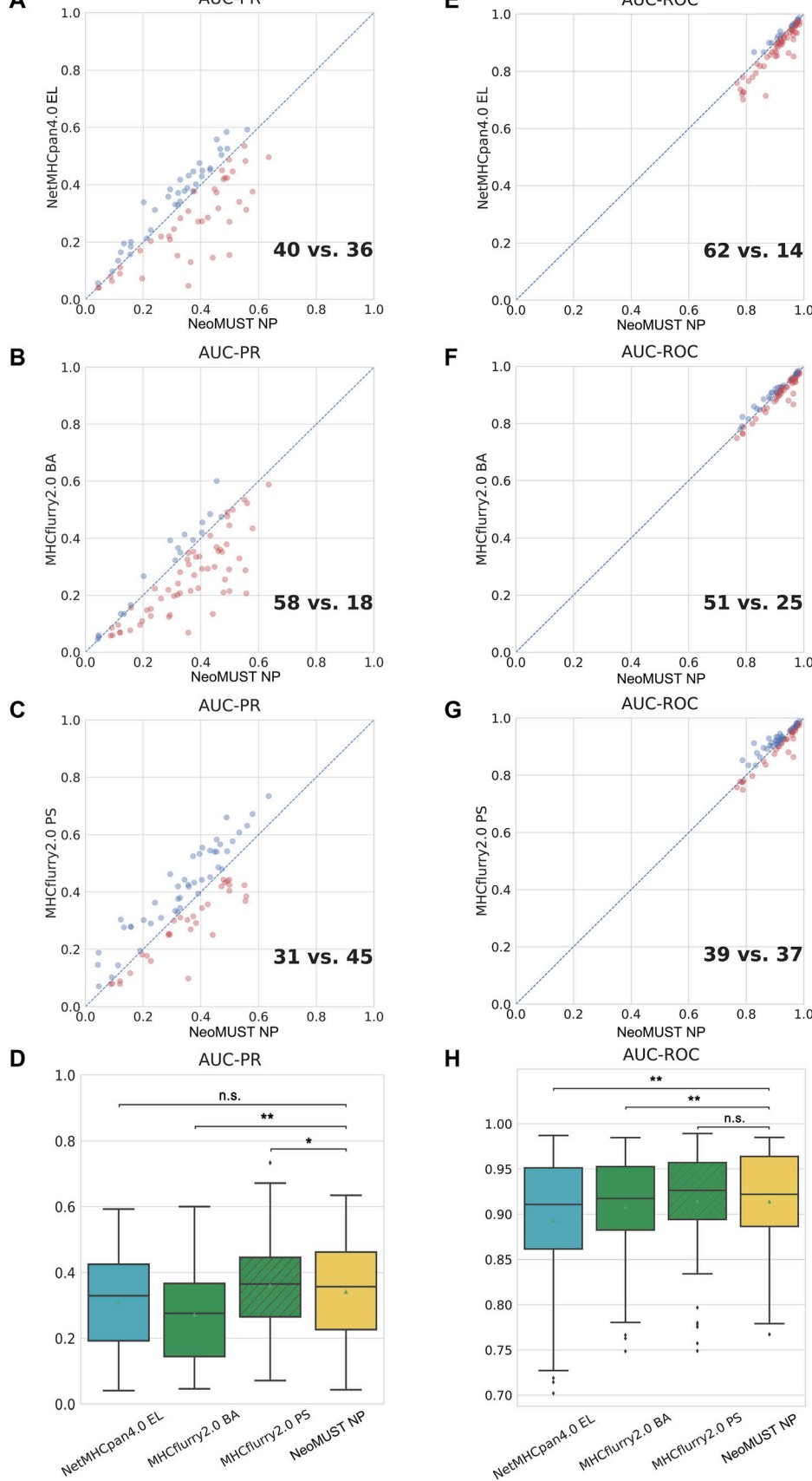

**Figure 2. NeoMUST exhibits a comparable performance on the multi-allelic test set, TeSet-1, compared with NetMHCpan4.0 and MHCflurry2.0.**

**(A, B, C)** Scatterplots of AUC-PRs of NeoMUST NP versus NetMHCpan4.0 EL, NeoMUST NP versus MHCflurry2.0 BA, and NeoMUST NP versus MHCflurry2.0 PS are illustrated in (A, B, C), respectively. **(E, F, G)** Scatterplots of AUC-ROCs of NeoMUST NP versus NetMHCpan4.0 EL, NeoMUST NP versus MHCflurry2.0 BA, and NeoMUST NP versus MHCflurry2.0 PS are illustrated in (E, F, G), respectively. Each data point represents each of total 76 samples in TeSet-1. Points in red signify samples demonstrating higher AUC-PR or AUC-ROC values for NeoMUST NP compared with alternative methods, whereas blue points denote instances of lower values for NeoMUST NP in these metrics. **(D, H)** Corresponding boxplots of AUC-PRs and of AUC-ROCs for TeSet-1 among NetMHCpan4.0 EL, MHCflurry2.0 BA, MHCflurry2.0 PS, and NeoMUST NP are shown in (D, H), respectively. In (D, H), ** indicates $P < 0.01$, * indicates $P < 0.05$, and n.s. represents being not significant. $P$-values were calculated using the Wilcoxon signed-rank test (see detailed statistics in Table S1).

Source data are available for this figure.

performance of predicting both positive and negative cases, NeoMUST NP exhibited a greater number of samples with the larger area under the receiver operating characteristic curves (AUC-ROCs) compared with those from NetMHCpan4.0 EL (62 versus 14 in Fig 2E), MHCflurry2.0 BA (51 versus 25 in Fig 2F), and MHCflurry2.0 PS (39 versus 37 in Fig 2G). The median (0.922) of AUC-ROCs of NeoMUST NP was significantly higher than that of NetMHCpan4.0 EL (median = 0.911, $P < 0.001$) and MHCflurry2.0 BA (median = 0.917, $P = 0.002$) and was comparable to that of MHCflurry2.0 PS (median = 0.926, $P = 0.860$), shown in Fig 2H. For a fair comparison, we conducted analyses specifically on common alleles trained by all models. The results, depicted in Fig S2A–C, were consistent with the aforementioned findings. It is worthy to note that owing to a significant data leakage issue, our model could not be directly compared with NetMHCpan4.1 and MixMHCpred2.2 on TeSet-1. Detailed comparison results, post-removal of the leaking data (67.12%), can be found in Fig S3A–C. In addition, we further broke down the performance metrics by various lengths of peptides in Fig S4A–C and by various alleles in Tables S2, S3, and S4. The results indicated that NeoMUST exhibited favorable predictive performance compared with state-of-the-art algorithms for neoantigen presentation.

NeoMUST also showed significantly improved performance in a mono-allelic test set of MS data, TeSet-2 (O'Donnell et al, 2020), which included 13, 271,500 peptide–MHC-I pairs with 1% true positives from 100 MS experiments using genetically engineered cell lines expressing a single MHC-I molecule. NeoMUST NP achieved a greater number of samples with larger AUC-PRs (62 versus 38 and 61 versus 39 in Fig 3A and B) and AUC-ROCs (75 versus 25 and 64 versus 36 in Fig 3D and E), respectively. Overall, NeoMUST NP demonstrated higher AUC-PRs (median = 0.637) compared with NetMHCpan4.0 EL (median = 0.631, $P = 0.065$) and MHCflurry2.0 BA (median = 0.630, $P = 0.050$), shown in Fig 3C. This trend persisted in PPV 1%, as evidenced in Fig S5A–C. Moreover, NeoMUST NP exhibited significantly elevated AUC-ROCs (median = 0.975) compared with NetMHCpan4.0 EL (median = 0.971, $P < 0.001$) and MHCflurry2.0 BA (median = 0.973, $P = 0.001$), depicted in Fig 3F. These results highlight the improved predictive accuracy of NeoMUST for both positive and negative cases. Fig S6A–C illustrated the comparison outcomes with NetMHCpan4.1 and MixMHCpred2.2 after removing 88.63% of the data in TeSet-2 to address the data leakage issue. Notably, MHCflurry2.0 PS was not included in the evaluation because its training data completely overlapped with TeSet-2 (O'Donnell et al, 2020) (see detailed statistics in Table S1 and performance metrics among MHC-I alleles in Tables S5, S6, and S7). In addition, NeoMUST demonstrated a comparable performance for our auxiliary task of predicting binding affinity for a recently archived (after 2020) dataset, TeSet-3. The Spearman correlation coefficients between the predicted scores versus the ground-truth values were 0.61, 0.60, and 0.61 for NetMHCpan4.1, MHCflurry2.0 BA, and NeoMUST BA, respectively, shown in Fig S7.

Moreover, NeoMUST demonstrated significantly reduced training time, especially when compared to ensemble models such as MHCflurry2.0 (Fig 4A), achieving a noteworthy time reduction of over 200-fold. Specifically, using our in-house DELL server equipped with two Intel Xeon CPUs (8 cores at 3.2 GHz), two NVIDIA GeForce RTX 3090 GPUs, and 256 GB of RAM, NeoMUST was successfully trained on the entire training set in 4.3 h. In contrast, MHCflurry2.0, with default training parameters, required 49.9 h for training on a mere 5% subset of the data. In our efforts to further optimize predictive performance, we introduced a NeoMUST ensemble model. This ensemble exhibited improved performance, achieving a higher AUC-PR compared with the single NeoMUST NP model on TeSet-1 (median of 0.397 versus 0.357, with a $P$-value < 0.001 in Fig 4B). Notably, the training time of the NeoMUST ensemble model remained 20-fold faster than that of MHCflurry2.0 (Fig 4A). This highlighted the computational efficiency of NeoMUST, leading to substantial time savings in model training and enhancing its scalability for neoantigen prediction (see the detailed comparison between single NeoMUST and NeoMUST ensemble models in Fig S8A–C).

At last, in elucidating the contributions of the MTL model architecture to effective learning and prediction, we conducted three analyses. Firstly, incorporating the auxiliary task of BA significantly enhanced the primary task of predicting neoantigen presentation. This was demonstrated by training a NeoMUST model while freezing the parameters of the BA expert layers and prediction tower, referred to as the NeoMUST-Drop-BA model. A comparison with the standard NeoMUST model on TeSet-1 revealed enhanced prediction performance of NeoMUST, with significantly larger AUC-PRs (median of 0.357 versus 0.317, $P < 0.001$ in Fig 5A) and AUC-ROCs (median of 0.922 versus 0.917, $P < 0.001$ in Fig S9), affirming the efficacy of incorporating a relevant auxiliary task.

Secondly, to assess the relative contributions of task-specific experts and the shared expert (Fig 1) for prediction, we analyzed the weights at the gates (Fig 1) representing the information flow into the prediction layers (Goodfellow, 2016). Results in Fig 5B revealed a significantly higher contribution from the shared expert compared with the task-specific expert for the BA prediction task (median weight of 0.751 versus 0.249, $P < 0.001$). This may be attributed to the small portion (20%) of the training data represented by BA, with the shared expert providing a substantial amount of representative information for the prediction of BA. Conversely, for the prediction of neoantigen presentation, the contribution was relatively balanced, with moderately higher weights from the task-specific expert (median of 0.564 versus 0.436, $P < 0.001$).

Thirdly, we ruled out the possibility that the improved prediction performance over MHCflurry2.0 BA was due to high sequence similarity between the NeoMUST training and test datasets. A direct comparison of sequence similarity between the training and test datasets showed that the average similarity between the NeoMUST training set and TeSet-1 was lower than that of MHCflurry2.0 BA ($P < 0.001$ in Fig 5C), eliminating the influence of sequence similarity on the observed improvements. Overall, these results underscored the effectiveness of the MTL architecture in enhancing prediction performance, independent of differences in training data.

## Discussion

Here, we presented NeoMUST, a deep learning model using an MTL approach to accurately predict neoantigen presentation through

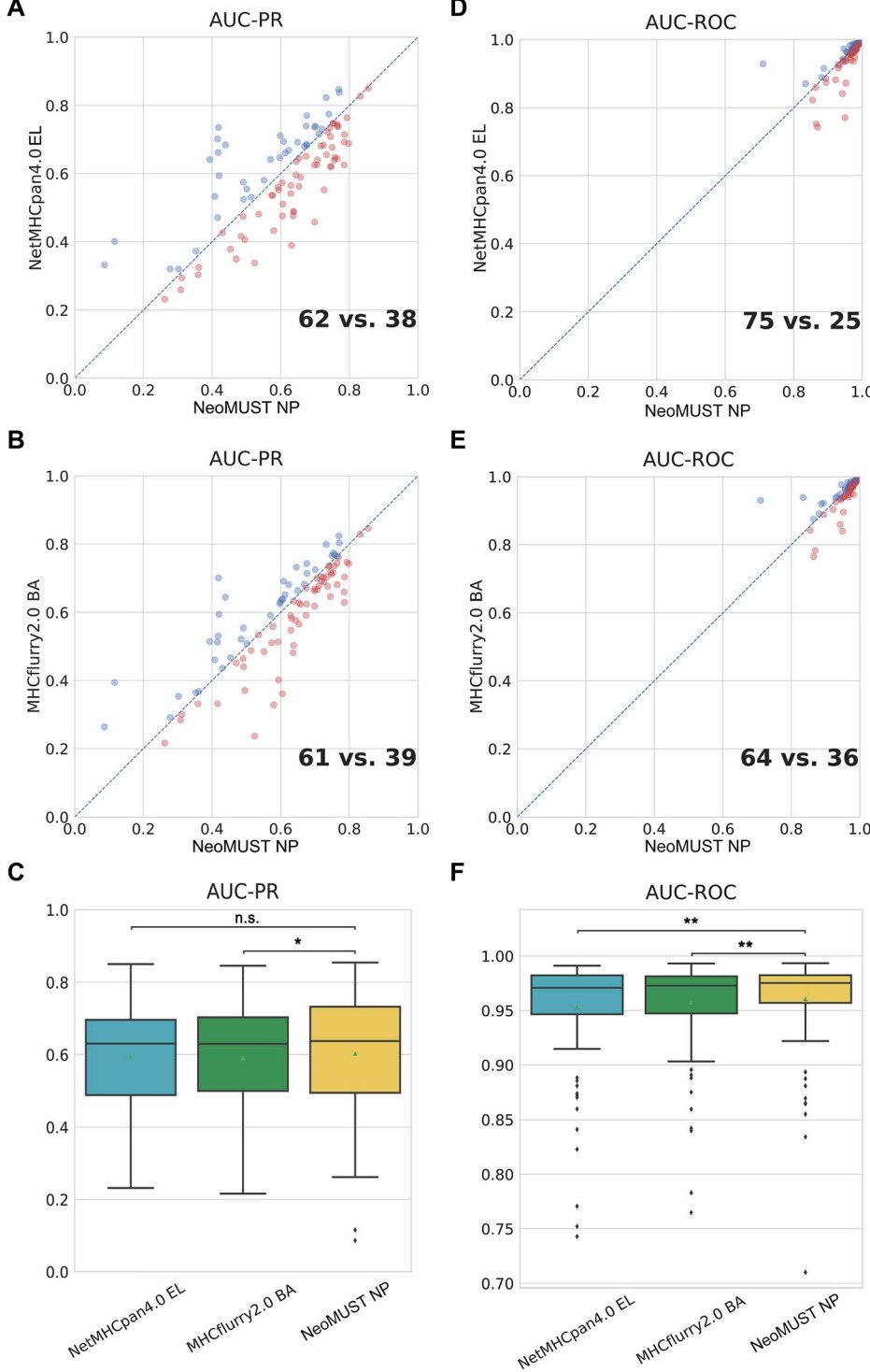

**Figure 3. NeoMUST exhibits a similar performance on the mono-allelic test set, TeSet-2, compared with NetMHCpan4.0 and MHCflurry2.0.**
**(A, B)** Scatterplots of AUC-PRs of NeoMUST NP versus NetMHCpan4.0 EL and NeoMUST NP versus MHCflurry2.0 BA are illustrated in (A, B).
**(D, E)** Scatterplots of AUC-ROCs of NeoMUST NP versus NetMHCpan4.0 EL and NeoMUST NP versus MHCflurry2.0 BA are illustrated in (D, E). Each data point corresponds to each of total 100 samples in TeSet-2. Red points indicate samples with superior AUC-PR or AUC-ROC values for NeoMUST NP compared with alternative methods, whereas blue points represent cases where NeoMUST NP exhibits lower values in these metrics. **(C, F)** Corresponding boxplots of AUC-PRs and of AUC-ROCs for TeSet-2 among NetMHCpan4.0 EL, MHCflurry2.0 BA, and NeoMUST NP are shown in (C, F), respectively. In (C, F), ** indicates $P < 0.01$, * indicates $P < 0.05$, and n.s. represents being not significant. $P$-values were calculated using the Wilcoxon signed-rank test (see detailed statistics in Table S1).
Source data are available for this figure.

MHC-I molecules. NeoMUST enables expert models to specialize in specific subsets of data, such as BA and MS data, to perform distinct prediction tasks, specifically, peptide–MHC-I binding and neo-antigen presentation. Moreover, our model uses a gating mechanism to adaptively learn the weights of the expert models, thereby facilitating effective handling of diverse types of training data and

prediction tasks (Tang et al, 2020). In contrast, ensemble models, including NetMHCpan4.1 and MHCflurry2.0, and other deep learning architectures such as recurrent neural networks and convolutional neural networks typically employ the same architecture for all input data, which may not be optimal for different data distributions and task characteristics (Ruder, 2017 *Preprint*; Thung & Wee, 2018).

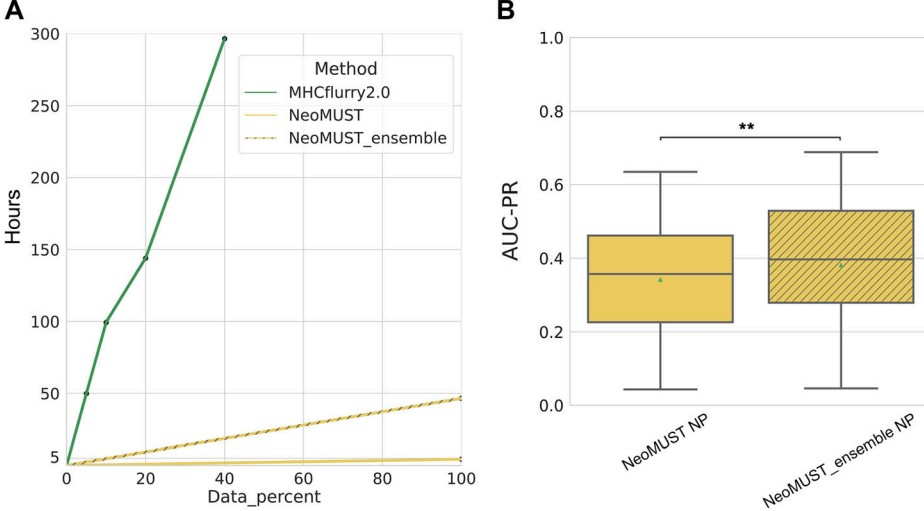

**Figure 4. Comparison of training time between MHCflurry2.0 BA, NeoMUST, and NeoMUST ensemble models.**
**(A)** Comparison of training time between MHCflurry2.0, NeoMUST, and NeoMUST ensemble models is shown in (A), using our in-house DELL server with two Intel Xeon CPUs and two NVIDIA GeForce RTX 3090 GPUs. **(B)** Boxplots of AUC-PRs comparing the prediction performance between NeoMUST NP and NeoMUST ensemble NP models are shown in (B). Each data point represents each of total 76 samples in TeSet-1. In (B), ** indicates *P* < 0.01. *P*-values were calculated using the Wilcoxon signed-rank test. Source data are available for this figure.

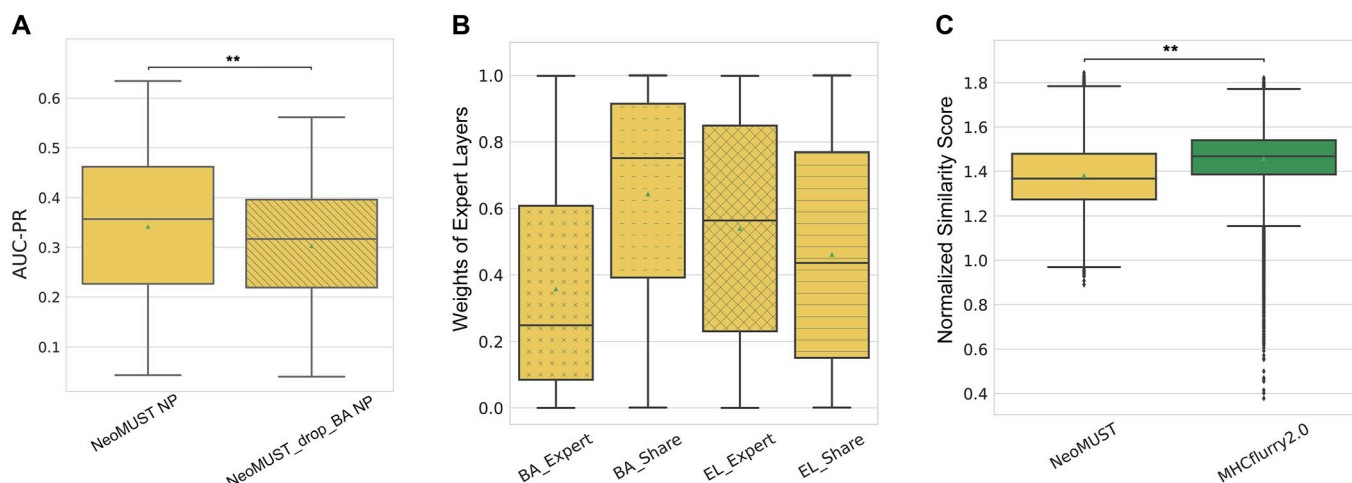

**Figure 5. Analyses of the model architecture.**
**(A)** Boxplots depicting AUC-PRs on TeSet-1 show the performance contrast between NeoMUST NP and NeoMUST-Drop-BA NP models in (A). Each data point represents each of total 76 samples in TeSet-1. **(B)** Boxplots illustrate weights at gates, highlighting the relative contributions of task-specific and shared experts in (B). **(C)** Boxplots illustrate sequence similarity comparisons between the training data of NeoMUST and the test data of TeSet-1 and between those of MHCflurry2.0 and TeSet-1 in (C). In (A, C), ** indicates *P* < 0.01. *P*-values in (A, B) were calculated using the Wilcoxon signed-rank test and *P*-value in (C) was calculated with the Wilcoxon rank-sum test. Source data are available for this figure.

In addition, to effectively balance the learning impact of the two tasks, we incorporated the CAGrad algorithm to optimize the gradient update direction, alongside the uncertainty weight algorithm to optimize the loss weights of these tasks. The prediction of neoantigen presentation involved a binary classification task, whereas the prediction of neoantigen–MHC binding entailed a regression task. Given the potential disparity in loss magnitudes between these tasks, the training process could be dominated by the task with the larger loss. (Cipolla et al, 2018). To address this, the CAGrad algorithm constructs new gradient update directions by determining gradient update directions within the vicinity of the mean gradient direction (Liu et al, 2021). In aggregate, the predictive accuracy of our model aligns with state-of-the-art algorithms. NeoMUST NP surpasses NetMHCpan4.0 EL, NetMHCpan4.1 EL,

MixMHCpred2.2, and MHCflurry2.0 BA, while demonstrating comparable performance to the MHCflurry2.0 PS model on the benchmark dataset TeSet-1, a representation of real-world clinical data, by employing AUC-ROC as the performance metric (Figs 2H and S3A). AUC-ROC provides a comprehensive measure balancing both positive and negative predictions. Our model exhibits comparable performance in terms of AUC-PR across all models, except for MHCflurry2.0 PS and MixMHCpred2.2, where our model displays marginally inferior results (Figs 2D and S3C). This observation is further supported by PPV 1% (Figs S3B and S1D).

Furthermore, the NeoMUST algorithm demonstrates enhanced scalability when compared to ensemble models such as NetMHCpan4.1 and MHCflurry2.0. These ensemble models comprise 40 neural network models (Jurtz et al, 2017) and 140 neural network

models (O'Donnell et al, 2020), respectively. In contrast, NeoMUST achieves a substantial reduction in training time, ~200-fold, while using the same computational resources.

The immunogenicity of neoantigens is governed by their presentation and recognition features (Wells et al, 2020). Neoantigen presentation, occurring at an early stage, is a requisite but not solely sufficient condition for immunity and effective tumor control (Martin et al, 2016; Rech et al, 2018; Ebrahimi-Nik et al, 2019; Brennick et al, 2021). Recognition features encompass a foreignness score, representing TCR recognition probability derived from homology to known pathogenic peptides (Richman et al, 2019), and agretopicity, defined as the ratio of mutant binding affinity to wild-type binding affinity (Duan et al, 2014). The applications of differential agretopicity indices have not only aided in identifying immunogenic neoepitopes (Ebrahimi-Nik et al, 2019) but also been demonstrated to significantly correlate with overall survival in advanced non–small-cell lung cancer (Ghorani et al, 2018). Given the scarcity of experimentally determined immunogenicity data, this study focused on neoantigen presentation. Nonetheless, we conducted tests on a small dataset, revealing that a two-step procedure combining predictors of neoantigen presentation, including NeoMUST, with agretopicity can enhance the identification of neoantigen immunogenicity (refer to detailed statistics in Fig S10).

Precise computational prediction of neoantigens is crucial in immunotherapy. The in silico identification and subsequent filtering strategy, followed by experimental validation, is a commonly employed methodology in various clinical studies. (Forghanifard et al, 2014; Ott et al, 2017; Cafri et al, 2020; Hu et al, 2021; Storkus et al, 2021). Our study introduces the NeoMUST model, adept at extracting task-specific and shared features from MS and BA data. This model enhances prediction performance compared with existing methods and substantially reduces training time. We anticipate that NeoMUST will contribute to the development of cancer immunotherapies.

# Materials and Methods

## Model architecture

We applied a customized gate control architecture (Fig 1). It improved learning performance for each task simultaneously and alleviated negative transfer, where prediction performance is weakened by conflicted tasks, compared with other advanced MTL architectures including hard parameter sharing models and a mixture of expert models (Tang et al, 2020).

The NeoMUST algorithm has three layers. First, *the embedding layer* (blue area in Fig 1) converts peptide sequences and MHC-I pseudo-sequences (Nielsen et al, 2007) into high-dimensional vectors, which are processed by bidirectional LSTM networks to extract sequence features. Next, in *the expert layer* (yellow area in Fig 1), the merged feature matrices are then fed into three task-specific expert neural networks with shared information merged through gate networks. Finally, *the prediction layer* (red area in Fig 1) contains two multi-layer perceptron (MLP) networks for classification of neoantigen presentation and prediction of binding

affinity, with binary cross-entropy and mean squared error serving as loss functions.

Specifically, at *the embedding layer*, the representation of peptides and alleles followed the methods in O'Donnell et al (2020). Briefly, a 21-dimensional vector was generated for each amino acid in the peptides using the BLOSUM62 substitution matrix (Henikoff & Henikoff, 1992). We encoded 11,609 MHC-I allele molecules as pseudo-sequences, using the 37 peptide-contacting positions as described by Nielsen et al (2007), Jurtz et al (2017), and O'Donnell et al (2020). Consequently, our model is a pan-allele model. Next, peptides were transformed into a 21 × 45 matrix, whereas MHC-I pseudo-sequences were transformed into a 21 × 37 matrix. NeoMUST employed two bilayer bidirectional long short-term memory (BiLSTM) neural networks, each with a hidden layer size of 32, to extract sequence features from the two matrices. The 64-dimensional feature matrices from the peptide and allele were obtained by taking all hidden layers in the corresponding BiLSTMs. These two feature matrices were then concatenated to form a 128-dimensional feature matrix.

Next, at *the expert layer*, the feature matrix from the embedding layer was fed to three task-specific expert neural networks, expert A for neoantigen presentation, expert B for binding affinity, and a shared expert for common features of the two tasks. Each expert consisted of MLPs of a hidden layer with a size of 128 and an output layer with a size of 64. The shared information between the experts was merged through a gate network (gate A and gate B) that learned and determined the weights of the final sequence of 64-dimensional features for the specific task and the shared expert.

Finally, *the prediction layer* consisted of two individual MLPs of a hidden layer with a size of 32 for classification of neoantigen presentation status and prediction of binding affinities, respectively. For the prediction of neoantigen presentation, the final percentile rank scores were estimated from an ensemble of presentation probabilities predicted using a set of 8- to 12-mer random natural peptides (50,000 of each length) as in Jurtz et al (2017) and pre-calculated presentation probabilities of random peptides are available at https://github.com/Freshwind-Bioinformatics/NeoMUST/tree/main/Data/rank_database_common. For the prediction of binding affinity, the outputted values in a 0–1 scale were transformed to a nanomolar (nM) affinity by output value from Neural Network Model = $1 - log_{50000}(nMaffinity)$ as in O'Donnell et al (2020).

## Model training

Our primary training dataset, TrSet-1, was derived from the training set, Data S3, of O'Donnell et al (2020), excluding non-human data. It comprises 476,685 MS data (459,172 positive and 17,513 negative cases) and 186,997 BA data collected from IEDB (2018 version) and the SysteMHC Atlas project (Kim et al, 2014; Shao et al, 2018; Abelin et al, 2019; Vita et al, 2019; Sarkizova et al, 2020). To balance positive and negative cases, we augmented TrSet-1 with the synthesized peptides as negative MS training data. Specifically, we randomly synthesized peptides of length 8–15 amino acids from the pool by combining all peptides of the positive MS data and randomly paired them with MHC-I alleles. During each epoch of training, 440,768

synthetic negative MS data were generated. After 40 epochs, previously generated negative data were recycled.

For evaluation on the mono-allelic test set, TeSet-2, overlapping data with TeSet-2 were removed from TrSet-1 to construct TrSet-2. TrSet-2 comprises 335,135 MS data (317,622 positive and 17,513 negative cases) and 186,997 BA data. Similar to TrSet-1, 299,136 synthetic negative MS data points were generated during each training epoch.

During the training process, TrSet-1 and TrSet-2 were split into training (90%) and validation (10%) sets. We employed the AdamW optimizer (Loshchilov & Hutter, 2019 *Preprint*) with a batch size of 1,024, a learning rate of $1 \times 10^{-3}$, and a weight decay of $1 \times 10^{-2}$. After 80 epochs, the model converged, that is, the loss of the validation set stops decreasing. Convergence, defined as the validation set loss stabilization, was observed after 80 epochs for both datasets.

To learn two tasks effectively and enhance neoantigen presentation prediction, we optimized the NeoMUST model through dynamically adjusting the weights of each task in the loss functions. We assessed their uncertainty using the uncertainty weight algorithm (Cipolla et al, 2018). Furthermore, we optimized gradients during the training process using the Conflict-Averse Gradient descent (CAGrad) algorithm (Liu et al, 2021) to resolve conflicting gradients where gradients of different task objectives are not well aligned.

### NeoMUST ensemble model

To optimize predictive performance, we developed a NeoMUST ensemble model, integrating models with diverse parameter sizes to capture data features comprehensively. Specifically, in addition to the baseline NeoMUST model, nine supplementary models were trained by systematically varying the number of layers and neurons in expert and prediction layers (see detailed parameters in Table S9). The final ensemble of five models was selected using a stepwise approach based on a reduction in validation loss (Ganaie et al, 2022).

### Implementation

The NeoMUST model was implemented in Python (version 3.9.12). We provided an easy-to-use command line interface (see software instruction and example test date sets at https://github.com/Freshwind-Bioinformatics/NeoMUST).

### Benchmark test

#### Benchmark test sets
We used a comprehensive multi-allelic mass spectrometry (MS) dataset (Data S1) from O'Donnell et al (2020) as TeSet-1. It consists of 76 individuals with a total of 9,158,100 data (91,581 positive and 9,066,519 negative cases). Each peptide underwent deconvolution with up to six corresponding HLA alleles, maintaining a consistent hit label (0 or 1) for all alleles. In the prediction process on the multi-allelic TeSet-1, NeoMUST, along with other predictors, we determined the predicted outcome for each peptide by selecting the highest score among the six alleles. In addition, we used a mono-allelic MS dataset (Data S2) from O'Donnell et al (2020) as

TeSet-2. It consists of 100 samples with a total of 13,271,500 data (132,715 positive and 13,138,785 negative cases).

In order to test the prediction of binding affinity, we constructed the test set 3, TeSet-3. Specifically, we downloaded all available binding affinity (BA) data submitted to IEDB (Vita et al, 2019) after 2020 to avoid overlapping with the training data of our and other benchmark models. The filters include Linear Sequence (Epitope Structure), Include Positive Assays, Include Negative Assays, Class I (MHC Restriction Type), *Homo sapiens* (Host), and MHC binding assay (MHC Assays). TeSet-3 consists of 1,186 BA values.

#### Performance metrics
For the assessment of neoantigen presentation prediction, we applied AUC-ROC (area under the receiver operating characteristic curve) and AUC-PR (area under the precision–recall curve) as primary metrics. AUC-ROC quantifies the area under the receiver operating characteristic curve, evaluating overall performance across different threshold values by plotting the true-positive rate against the false-positive rate. AUC-PR calculates the area under the precision–recall curve, providing a comprehensive assessment of classifier performance, especially in scenarios with imbalanced class distributions. In addition, we computed the PPV, specifically PPV 1%, representing the ratio of true positives within the top 1% of predicted scores per sample. 1% was chosen to balance the large variance in the number of neoantigens among the individuals.

For the assessment of binding affinity prediction, we applied the Spearman correlation coefficients as metrics. Furthermore, *P*-values were calculated using the Wilcoxon signed-rank test to compare the performance metrics between any two predictors.

#### Comparison with NetMHCpan4.0 and MHCflurry2.0 BA and PS
We compared NeoMUST with the NetMHCpan4.0 (Jurtz et al, 2017) eluted ligand (EL) prediction model with the default parameters, and MHCflurry2.0 (O'Donnell et al, 2020) BA and PS models with no-flanking and default parameters for the peptide lengths of 8, 9, 10, and 11 on TeSet-1. In addition, we also compared NeoMUST with NetMHCpan4.0 and MHCflurry2.0 on TeSet-2 with the default parameters.

To compare the training time among single NeoMUST and ensemble models, we retrained the NeoMUST ensemble model and MHCflurry2.0 BA with various proportions, 5%, 10%, 20%, and 40%, of TrSet-1 using the default hyperparameters for MHCflurry2.0 BA using our in-house DELL server with two Intel Xeon CPUs (8 cores at 3.2 GHz), two NVIDIA GeForce RTX 3090 GPUs, and 256 GB of RAM.

#### Comparison with NetMHCpan4.1 and MixMHCpred2.2
To facilitate a comparison with NetMHCpan4.1 and MixMHCpred2.2, we mitigated potential data leakage from TeSet-1 and TeSet-2. Specifically, we excluded leaked positive neoantigens and individuals with less than 0.1% positive cases. This process resulted in TeSet-1-Filtered, comprising 25 individuals with a total of 3,011,436 data points (22,108 positive cases and 2,989,328 negative cases, accounting for 32.88% of TeSet-1). Similarly, TeSet-2-Filtered was established, encompassing 31 samples with a total of 1,509,188 data points (6,927 positive cases and 1,502,261 negative cases, constituting 11.37% of TeSet-2). Benchmark studies were then conducted

on TeSet-1-Filtered and TeSet-2-Filtered, respectively, using the default parameters of NetMHCpan4.1 and MixMHCpred2.2.

### Analysis of the NeoMUST model architecture

Firstly, we validated the enhancement of neoantigen presentation prediction by incorporating the auxiliary task of BA. This was achieved through training a NeoMUST-Drop-BA model, preserving the base model architecture while restricting parameter updates in the BA task-specific expert and BA prediction layers during back-propagation. Secondly, we investigated the specific contributions of task-specific and shared experts in prediction by applying the NeoMUST model to TeSet-1. We quantified expert contributions using their weights at each gate for individual cases (Ruder, 2017 Preprint), comparing these weights using a Wilcoxon signed-rank test.

Finally, to address potential confounding effects from differences in training datasets, we assessed the sequencing similarity between the MHCflurry2.0 BA training set and TeSet-1 and between the Neo-MUST training set and TeSet-1. MHCflurry2.0 was selected for comparison because of its minimal disparities in training data with NeoMUST. The positive cases between NeoMUST and the MHCflurry2.0 BA model were identical, with the primary distinction residing in the negative cases in training data. For sequence similarity assessment, 10,000 negative examples were randomly sampled without replacement from a total of 9,066,519 cases in TeSet-1. Subsequently, we randomly selected 10,000 negative examples from a pool of 342,797 cases in both MHCflurry2.0 and NeoMUST, respectively. This process was iterated 10 times, and the results were aggregated for subsequent analysis. The similarity scores were computed using the pairwise2.align.globalds function (Cock et al, 2009) from the Biopython library and normalized by their alignment lengths.

### Prediction of neoantigen immunogenicity

We obtained a dataset from Hu et al (2021), where the immuno-genicity of neoantigens was experimentally determined using the IFN-γ ELISpot assay. Subsequently, we filtered out individuals lacking positive cases, resulting in the creation of the test set, TeSet-4. This dataset comprises five individuals, encompassing 107 neoantigens, with 13 classified as positive cases and 94 as negative cases.

We assessed the accuracy of neoantigen immunogenicity prediction using PPV5 as the performance metric, representing the percentage of true positives among the top five predicted scores for neoantigens. The evaluation was conducted to compare the results from two procedures. The first procedure involved predicting neoantigen immunogenicity solely through neoantigen presentation algorithms. In the second procedure, following the two-step approach recommended by Wells et al (2020), neoantigens were initially ranked based on predicted scores from neoantigen prediction algorithms. Subsequently, those with an agretopicity value more than 0.1 were filtered out, and the resulting ranked neoantigens were quantitatively assessed using PPV5.

The use of benchmark models and dataset details can be referenced in Table S10.

## Data Availability

NeoMUST is implemented in Python. It is freely accessible at the GitHub repository (https://github.com/Freshwind-Bioinformatics/NeoMUST).

## Supplementary Information

## Acknowledgements

The authors express their gratitude to Drs Fuli Yu, Imtiaz Yakub, and Anindita Ravindran for their diligent review of the article and valuable feedback, which greatly contributed to the improvement of the quality of this work.

### Author Contributions

W Ma: conceptualization, data curation, software, formal analysis, validation, visualization, methodology, and writing—original draft, review, and editing.
J Zhang: data curation, formal analysis, validation, and writing—review and editing.
H Yao: conceptualization, resources, software, formal analysis, supervision, validation, investigation, methodology, and writing—original draft, review, and editing.

### Conflict of Interest Statement

The authors declare that they have no conflict of interest.

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
