## [Reviewer comments · Life Science Alliance]

Life Science Alliance

NeoMUST: an Accurate and Efficient Multi-Task Learning Model for Neoantigen Presentation

Wang Ma, Jiawei Zhang, and Hui Yao

DOI: <https://doi.org/10.26508/lsa.202302255>

Corresponding author(s): Hui Yao, Fresh Wind Biotechnologies USA Inc.

Review Timeline:

Submission Date:	2023-07-06
Editorial Decision:	2023-09-06
Revision Received:	2023-12-05
Editorial Decision:	2024-01-12
Revision Received:	2024-01-19
Accepted:	2024-01-22

Transaction Report:

September 6, 2023

Re: Life Science Alliance manuscript #LSA-2023-02255-T

Hui Yao
Fresh Wind Biotechnologies USA Inc.

Dear Dr. Yao,

Thank you for submitting your manuscript entitled "NeoMUST: an Accurate and Efficient Multi-Task Learning Model for Neoantigen Presentation" to Life Science Alliance. The manuscript was assessed by expert reviewers, whose comments are appended to this letter. We invite you to submit a revised manuscript addressing the Reviewer comments.

Thank you for this interesting contribution to Life Science Alliance. We are looking forward to receiving your revised manuscript.

Sincerely,

B. MANUSCRIPT ORGANIZATION AND FORMATTING:

Reviewer #1 (Comments to the Authors (Required)):

The authors presented a new tool for neoantigen presentation prediction, namely NeoMUST. The tool uses a different modeling approach than commonly used prediction tools such as NetMHCpan4.0 and MHCflurry2.0 by enabling expert models to specialize in specific types of training data. By comparing NeoMUST to existing tools, the authors were able to achieve a higher prediction accuracy using three different validation datasets (in AUC but more importantly PPV). On top of this, NeoMUST shows significant reduction in training time compared to MHCflurry2.0 when using the same computational resources.

Overall, the manuscript is well written and datasets were chosen reasonably. It was great to see the authors achieving similar if not better prediction accuracy in tested samples when using a different model architecture compared to current approaches. The reduction of training time is also a nice feature given the rapid expansion of available training data and the potential need to retrain algorithms with higher frequency. I have a couple comments below:

- When using multi-allelic MS data, how are positives being defined? Does NeoMUST take in all 6 class I alleles and give a combined output or are the authors considering a peptide to be predicted as positive if it can bind to (or be presented by) any of the 6 alleles? Was the multi-allelic MS data deconvoluted prior to training/testing?
- The authors highlight that NeoMUST is a predictor for HLA Class I peptide presentation. For a better comparison on that front, have the authors considered using NetMHCpan4.1 EL and the MHCflurry AP model (BA + PS) instead of the currently used NetMHCpan4.0 and MHCflurry BA only model?
- According to NeoMUST github page, NeoMUST has BA, EL as well as EL rank models. Can authors clarify which of these models were used in the model comparisons? Additionally, to further highlight the advantage of NeoMUST having a short training time, can NeoMUST be retrained by others using their own in-house datasets?

Reviewer #2 (Comments to the Authors (Required)):

The study demonstrates application of a deep learning approach to the problem of predicting peptide presentation by MHC-I, and predicting its binding affinity. There are several methods published on this topic, and this study focuses on demonstrating the use of multi-tasking to this problem. This is similar to what NetMHCpan does, but NeoMUST uses a different architecture of expert and shared models, besides making use of other advances in deep learning. It is not clear based on the results whether the improved performance is due to a multi-tasking approach or due to differences in training data. Comparisons were made to NetMHCpan and MHCflurry, but there are other published methods that were not included in the comparisons (eg HL Athena, MixMHCpred, SHERPA). Several important investigations of such a model were not performed.

Minor, specific comments

- It should be clarified upfront in the results section that the PPV is based on the top 1% scores, as it is an important immediate consideration.
- Figure 2: Please specify the statistical test used to compare the AUCs.

Overall comments on the study

- A multi-tasking approach was used, including binding affinity and elution as the outcomes, but the authors did not demonstrate that multi-tasking was beneficial for this problem. For example, does the performance of elution prediction degrade if the binding affinity task is dropped?
- Similarly, what was the benefit of using shared and expert models? Where was the major learning happening, in the shared or in the expert models?
- The size of the training data is expected to impact performance. While a fair comparison would be based on training the comparator models on the same training data, this may not be possible for some of the comparator models. Nevertheless, one can still have more informative comparisons: for example, separately comparing for the alleles that both models have been trained on, and alleles that NeoMUST is uniquely trained on, etc.
- This looks like a pan-allele method, but it is not discussed or demonstrated.
- The discussion on ensemble methods is geared towards computational resources. However, that is an expected cost of using ensemble methods, for the benefit of improved performance. It has to be demonstrated that ensemble methods did not benefit

NeoMUST.

Reviewer #3 (Comments to the Authors (Required)):

This is a superficial study which does not show any understanding of the nuances of the current literature and adds nothing to our knowledge. See specifics:

1. The study focuses on antigen presentation by MHC I molecules. It does not recognize the fact that most of the peptides that can actually mediate tumor control in vivo bind MHC I very poorly as seen in mouse models (Duan et al. 2014 J Exp Med, Brennick et al. 2021 JCI) and also in humans (Ghorani et al. 2018 Ann. Oncology, Rech et al. 2018 Cancer Immunology Res.).
2. The peptides that are predicted to bind MHC I very well actually do not work in tumor control (Martin et al. 2016 PLOS One, Brennick et al. 2021 JCI).
3. Even among the neoepitopes that are presented well by MHC I strongly, only a small minority can mediate tumor control (Ebrahimi-Nik et al. 2019 JCI Insight).
4. It is not clear how the authors show superior performance in Figs. 2 and 3. There are no experiments here to get to the gold truth. How do they show their algorithm is superior?

Referee cross-comments

I have read with interest and respect the other referees' comments. Since my expertise is different from theirs, I point out here that regardless of the technical issues (positive and negative) that the other referees point out, the manuscript is flawed in its essential premise - that identifying neoepitopes that are presented, are of some significant value. Even when neoepitopes have been actually identified experimentally by MS, they are mostly ineffective and MS misses most of the true anti-tumor neoepitopes anyway.

Reviewer: #1

1- When using multi-allelic MS data, how are positives being defined? Does NeoMUST take in all 6 class I alleles and give a combined output or are the authors considering a peptide to be predicted as positive if it can bind to (or be presented by) any of the 6 alleles? Was the multi-allelic MS data deconvoluted prior to training/testing?

Thank you for your thoughtful questions. When handling multi-allelic mass spectrometry (MS) data from TeSet-1, each peptide underwent deconvolution with up to six corresponding HLA alleles, maintaining a consistent hit label (0 or 1) for all alleles. During the prediction process on multi-allelic TeSet-1, NeoMUST, along with other benchmark predictors, determined the predicted outcome for each peptide by selecting the highest score among the six alleles.

Consequently, the deconvolution of multi-allele MS data was conducted prior to testing, while the training data of NeoMUST exclusively consisted of mono-allelic data. A detailed explanation of the deconvolution process for multi-allelic MS data was included in the Benchmark test subsection on page 16. We appreciate your attention to these details and hope this provides clarity regarding our methodology.

2- The authors highlight that NeoMUST is a predictor for HLA Class I peptide presentation.

For a better comparison on that front, have the authors considered using NetMHCpan4.1 EL and the MHCflurry AP model (BA + PS) instead of the currently used NetMHCpan4.0 and MHCflurry BA only model?

Thank you for the thoughtful question. We considered to compare with NetMHCpan4.1 EL and MHCflurry 2.0 PS model (BA+AP) but dropped them in the original manuscript due to

their data leakage issue to our benchmark data sets, TeSet-1 and TeSet-2. In the revised manuscript, NeoMUST was compared with MHCflurry 2.0 PS on TeSet-1, revealing comparable results for PPV and AUC-ROC metrics, shown in Figure 2 in RESULTS section. Due to training set overlap, MHCflurry 2.0 PS could not be tested on TeSet-2, as explained in the RESULTS section in page 8.

To facilitate a comparison with NetMHCpan4.1, we introduced filtered datasets, TeSet-1-Filtered and TeSet-2-Filtered, and assessed NeoMUST against NetMHCpan4.1 EL. Results, including Figure S5, Figure S6, and Supplementary Data 4, are detailed in the revised article. Notably, NeoMUST demonstrated better AUC-ROC performance compared to NetMHCpan4.1 EL on both filtered test sets. While exhibiting comparable PPV with NetMHCpan4.1 EL on TeSet-1-Filtered, NeoMUST showed a moderate decrease in PPV on TeSet-2-Filtered. In summary, NeoMUST's performance aligns with MHCflurry 2.0 PS and NetMHCpan4.1 EL across various evaluation metrics.

3- According to NeoMUST github page, NeoMUST has BA, EL as well as EL rank models. Can authors clarify which of these models were used in the model comparisons? Additionally, to further highlight the advantage of NeoMUST having a short training time, can NeoMUST be retrained by others using their own in-house datasets?

Thank you for your insightful inquiries. NeoMUST offers two models: BA and EL. The term "EL rank model" refers to the computation of percentile rank scores based on EL model results. All model comparisons were conducted using the EL percentile rank score, except for a benchmark test on the BA task depicted in Figure S2. A comprehensive description of EL

and BA predictions is available in Supplementary Data 2.3. We acknowledge any confusion in the GitHub page statements and have rectified them accordingly.

Moreover, we affirm that the NeoMUST training model has been made available on the GitHub repository

(https://github.com/Freshwind-Bioinformatics/NeoMUST/blob/main/Neomust/train_cl.py),

facilitating others to retrain the model with their respective datasets. We appreciate your attention to detail and value the opportunity to clarify these aspects.

Reviewer: #2

1- The study demonstrates application of a deep learning approach to the problem of predicting peptide presentation by MHC-I, and predicting its binding affinity. There are several methods published on this topic, and this study focuses on demonstrating the use of multi-tasking to this problem. This is similar to what NetMHCpan does, but NeoMUST uses a different architecture of expert and shared models, besides making use of other advances in deep learning. It is not clear based on the results whether the improved performance is due to a multi-tasking approach or due to differences in training data.

We appreciate your insightful question regarding the factors contributing to the performance improvement observed in our study. To isolate the impact of the multitasking approach employed in NeoMUST from potential differences in training data, we conducted a comparison with MHCflurry2.0, a method with minimal divergence in training data. To assess the dissimilarity in training data between NeoMUST and MHCflurry2.0, we measured sequence similarity between their training datasets and the test data, TeSet-1. The

results, depicted in Fig. 5C, indicated a significantly lower sequence similarity for NeoMUST and TeSet-1 (median = 1.367) compared to MHCflurry2.0 (median = 1.468, $p = 0$). Further details can be found in the last paragraph of the RESULTS section on page 10 and Supplementary Data 5.

This disparity implies that NeoMUST predicts TeSet-1 to be more challenging than MHCflurry2.0 solely based on training data considerations. Consequently, our findings suggest that the enhanced performance of NeoMUST can be attributed to the utilization of a multitasking approach, independent of training data variations. We sincerely appreciate your thoughtful consideration, which greatly contributes to our efforts in providing a clear understanding of our methodology.

2- Comparisons were made to NetMHCpan and MHCflurry, but there are other published methods that were not included in the comparisons (eg HLAthena, MixMHCpred, SHERPA). We appreciate the insightful query regarding the comparison of NeoMUST with other prediction methods. Unfortunately, both HLAthena and SHERPA were inaccessible for testing purposes. HLAthena's web-based server could not handle our large benchmark sets, such as TeSet-1, and its docker was not accessible to our team. Similarly, SHERPA, being proprietary software, was unavailable for testing.

While MixMHCpred2.2 was considered for comparison, its data leakage issue to our benchmark datasets, TeSet-1 and TeSet-2, led to its exclusion from the original manuscript. In the revised version, we addressed this concern by introducing filtered datasets, namely TeSet-1-Filtered and TeSet-2-Filtered, enabling a comparative assessment of NeoMUST

against MixMHCpred2.2. Detailed results, including Figures S5 and S6, as well as Supplementary Data 4, were provided in the revised article. Notably, NeoMUST exhibited better AUC-ROC performance with a moderate decrease in PPV on the filtered datasets compared to MixMHCpred2.2.

3- A multi-tasking approach was used, including binding affinity and elution as the outcomes, but the authors did not demonstrate that multi-tasking was beneficial for this problem. For example, does the performance of elution prediction degrade if the binding affinity task is dropped?

We appreciate your insightful question. In response, we conducted the suggested analysis by training a NeoMUST model while freezing the parameters of the BA expert layers and prediction tower, resulting in the NeoMUST-Drop-BA model. Comparative assessment with the standard NeoMUST model on TeSet-1 indeed revealed a diminished predictive performance in the NeoMUST-Drop-BA model, evidenced by lower Positive Predictive Values (PPVs) (median of 0.413 versus 0.439, $p < 0.001$) and smaller Area Under the Receiver Operating Characteristic (AUC-ROC) values (median of 0.917 versus 0.922, $p < 0.001$). This underscored the effectiveness of incorporating the relevant auxiliary task. Further elaboration on this analysis was provided in the first paragraph of Page 10 and illustrated in Figure 5A.

We are grateful for your suggestions, contributing to a clearer understanding of the benefits of the multi-task learning architecture.

4- Similarly, what was the benefit of using shared and expert models? Where was the major learning happening, in the shared or in the expert models?

Thank you for your insightful inquiry. In response to assessing the impact of shared and task-specific expert models on prediction, we conducted an analysis of the weights at the gates governing information flow into the prediction layers. Our findings indicated a significantly higher contribution from the shared expert compared to the task-specific expert for the binding affinity (BA) prediction task (median weight of 0.751 versus 0.249, $p < 0.001$). This observation was attributed to the relatively small portion (20%) of training data represented by BA, with the shared expert providing substantial representative information crucial for BA prediction. Conversely, for neoantigen presentation prediction, the contribution was more balanced, with moderately higher weights from the task-specific expert (median of 0.564 versus 0.436, $p < 0.001$). Detailed explanations of this analysis were provided in the second paragraph on page 10 and illustrated in Figure 5B. We appreciate your suggestion, which has enabled us to offer clarity on the model's mechanism.

5- The size of the training data is expected to impact performance. While a fair comparison would be based on training the comparator models on the same training data, this may not be possible for some of the comparator models. Nevertheless, one can still have more informative comparisons: for example, separately comparing for the alleles that both models have been trained on, and alleles that NeoMUST is uniquely trained on, etc.

We appreciate your insightful question. To investigate the impact of the differences in training data on prediction performance, we pursued two avenues of analysis. First, we clarified that sequence similarity between training and test sets does not singularly dictate prediction performance, as detailed in the response to your first question shown as above. Additionally,

we addressed your suggestion by conducting predictions specifically on the common alleles present in TeSet-1, which all models, including MetMHCpan4.0, MHCflurry2.0 BA, MHCflurry2.0 PS, and NeoMUST, were trained on. The results, presented in Figure S4, consistently mirrored the overall performance trends illustrated in Figure 2. These findings affirmed that prediction improvement is not primarily determined by differences in the training set. We value your guidance, and these analyses provide further insights into the robustness of our model across shared alleles.

6- This looks like a pan-allele method, but it is not discussed or demonstrated.

Thank you for your inquiry. Our approach involved representing MHC-I alleles as pseudo-sequences, aligning with the methodologies outlined by O'Donnell et al. (2020) and Nielsen et al. (2007), thereby rendering our model a pan-allele method. We incorporated additional explanations in Supplementary Data 2.1. We appreciate your attention to detail, which has assisted us in enhancing the clarity of our model description.

7- The discussion on ensemble methods is geared towards computational resources.

However, that is an expected cost of using ensemble methods, for the benefit of improved performance. It has to be demonstrated that ensemble methods did not benefit NeoMUST.

We appreciate your insightful suggestion. We introduced a NeoMUST ensemble model. The ensemble demonstrated enhanced performance in achieving a higher PPV compared to the single NeoMUST model (median of 0.454 versus 0.439, as illustrated in Figure 4B). Importantly, despite the enhanced performance of the NeoMUST ensemble model, it is

noteworthy that the training time for this model remained 20-fold faster than that of MHCflurry 2.0, as illustrated in Figure 4A. We incorporated a detailed explanation of the NeoMUST ensemble model and its performance in the first paragraph of Page 9 in the RESULTS section. We maintain that NeoMUST remains an efficient and effective model, preserving its inherent advantage of shorter training time, even when training and utilizing its ensemble counterpart.

8- It should be clarified upfront in the results section that the PPV is based on the top 1% scores, as it is an important immediate consideration

We appreciate your valuable feedback. We have incorporated the suggested clarification regarding PPV in the the first paragraph of the RESULTS section.

9- Figure 2: Please specify the statistical test used to compare the AUCs.

Thank you for your insightful comment. We have addressed your suggestion by including a statement in the Figure 2 caption specifying the use of the Wilcoxon signed-rank test for comparing performance metrics between any two predictors.

Reviewer: #3

1- The study focuses on antigen presentation by MHCI molecules. It does not recognize the fact that most of the peptides that can actually mediate tumor control in vivo bind MHCI very poorly as seen in mouse models (Duan et al. 2014 J Exp Med, Brennick et al. 2021 JCI) and also in humans (Ghorani et al. 2018 Ann. Oncology, Rech et al. 2018 Cancer Immunology

Res.). 2. The peptides that are predicted to bind MHC I very well actually do not work in tumor control (Martin et al. 2016 PLOS One, Brennick et al. 2021 JCI). 3. Even among the neoepitopes that are presented well by MHC I strongly, only a small minority can mediate tumor control (Ebrahimi-Nik et al. 2019 JCI Insight).

Thank you for highlighting the crucial aspect of neoantigen presentation and providing valuable references. We acknowledge that neoantigen presentation, while a necessary early step, is not solely determinant of immunogenicity and effective tumor control. In response to your feedback, we have reinforced our manuscript in the following ways:

- 1) Explicitly stated that neoantigen presentation is a requisite but not solely sufficient condition for immunity and effective tumor control, as indicated by the referenced studies (Duan et al. 2014, Brennick et al. 2021, Ghorani et al. 2018, Rech et al. 2018, Martin et al. 2016, Ebrahimi-Nik et al. 2019) at the second paragraph on Page 13 in DISCUSSION section.
- 2) Incorporated references (Duan et al. 2014, Ebrahimi-Nik et al. 2019) to underscore the importance of neoantigen recognition features, particularly agretopicity at the second paragraph on Page 13 in DISCUSSION section.
- 3) Acknowledged the limited availability of experimentally determined immunogenicity data and clarified that our study primarily focuses on neoantigen presentation. However, we conducted tests on a small dataset, demonstrating that a two-step procedure combining predictors of neoantigen presentation, including NeoMUST, and agretopicity, can enhance the identification of neoantigen immunogenicity. Detailed statistics are provided in Figure S8, and test methods are outlined in Supplementary

Data 6.

We hope that these updates address your concerns and contribute to the clarity of our manuscript."

2- It is not clear how the authors show superior performance in Figs. 2 and 3. There are no experiments here to get to the gold truth. How do they show their algorithm is superior?

Thank you for your thoughtful inquiry. We adjusted our language to reflect that our model demonstrates comparable prediction performance with the state-of-the-art model, rather than claiming superiority. Our conclusion was drawn from testing on a substantial benchmark test set, aligning with common practices observed in studies related to the development of computational models for cancer immunotherapy, exemplified by O'Donnell et al. 2020 (MHCflurry2.0) and Reynisson et al. 2020 (NetMHCpan4.1).

While acknowledging the significance of experimental validation as a gold standard, regrettably, it lies outside the scope of our current study. Our principal aim is to provide an effective and computationally efficient tool designed for researchers interested in predicting neoantigen presentation, especially those with constraints on computational resources and seeking to train their own in-house data. We appreciate your understanding of the study's focus and objectives.

3- I have read with interest and respect the other referees' comments. Since my expertise is different from theirs, I point out here that regardless of the technical issues (positive and negative) that the other referees point out, the manuscript is flawed in its essential premise -

that identifying neoepitopes that are presented, are of some significant value. Even when neoepitopes have been actually identified experimentally by MS, they are mostly ineffective and MS misses most of the true anti-tumor neoepitopes anyway

We appreciate your opinion about the prediction of neoantigen presentation. As of our current understanding, the accurate identification of neoantigen immunogenicity on a large scale remains an active area of research. A prevalent strategy involves the computational prediction of neoantigen presentation, followed by the filtration of resulting candidates using neoantigen expression data and recognition features such as agretopicity, with experimental validation as the concluding step. This approach has been widely employed in numerous clinical studies. For instance, netMHCpan 3.0 was utilized in a clinical trial (NCT03480152) for predicting neoantigens in mRNA vaccine development targeting gastrointestinal cancers (Cafri G, et al. JCI 2020). Similarly, netMHCpan 2.4 was employed in a clinical trial (NCT01970358) to predict neoantigens, subsequently informing the construction of personalized neoantigen vaccines for patients with melanoma (Ott PA. et al. Nature 2017). Prominent tools like NetMHCpan series models, utilized in these studies, inherently are predictors of neoantigen presentation, leveraging the wealth of available neoantigen presentation data, including mass spectrometry (MS) data. As elucidated in our responses to prior queries, we have aligned with this established procedure by filtering top predicted candidates based on recognition features. We assert that the prediction of neoantigens remains a pertinent and valuable pursuit in cancer immunotherapy, and NeoMUST serves as a valuable tool for advancing this objective.

January 12, 2024

RE: Life Science Alliance Manuscript #LSA-2023-02255-TR

Dr. Hui Yao
Fresh Wind Biotechnologies USA Inc.
4502 Riverstone Blvd, STE1104
Missouri City, Texas 77459

Dear Dr. Yao,

Thank you for submitting your revised manuscript entitled "NeoMUST: an Accurate and Efficient Multi-Task Learning Model for Neoantigen Presentation". We would be happy to publish your paper in Life Science Alliance pending final revisions necessary to meet our formatting guidelines.

- please address Reviewer 2's minor comments
- please be sure that the authorship listing and order is correct
- please add callouts for Figures S3A-C; S4A-B; S6A-B to your main manuscript text
- the Supplemental Data should be incorporated into the Materials and Methods section of the main manuscript. The References listed in the Supplemental file should be incorporated into the main Reference list.

A. FINAL FILES:

B. MANUSCRIPT ORGANIZATION AND FORMATTING:

Sincerely,

Reviewer #1 (Comments to the Authors (Required)):

My comments/questions have been addressed in the revised manuscript.

Referee Cross-Comments

I agree with Reviewer 2 that the paper highlights a different approach to a challenge the field has been actively working on but with limited increase in performance compared to other popular algorithms.

Reviewer #2 (Comments to the Authors (Required)):

I appreciate the analyses conducted by the authors to address the questions raised earlier, and including comparisons to MixMHCpred. I think this study adds utility to the field in terms of highlighting how a multi-task learning approach can be applied to combine peptide-MHC binding affinity and presentation data, contrasting with the fully-shared-network approach taken by NetMHCpan for this question. However, the performance seems to be only at a similar level to newer methods like MixMHCpred2.0 or NetMHCpan-4.1. The authors were not able to compare with several other recent methods, which makes it difficult to understand if there are performance gains. Overall, I think this study provides an incremental conceptual development about how deep learning approaches can tackle multiple but related outcome data for the purpose of peptide-MHC presentation prediction.

Other minor comments:

- In Fig 1, it's not clear what output of NetMHCpan is compared to, BA or EL? And similarly, what output of NeoMUST - presentation or affinity?
- The figure legends should clearly indicate what the different data points in the scatter plots and boxplots represent (samples from 76 individuals?)
- Older methods like NetMHCpan-4.0 were not trained on as much presentation data, so why not just use the newer versions?
- Instead of PPV at 1%, which is a particular snapshot of a precision-recall curve, the authors should use AUPRC as the performance metric.

We extend our sincere gratitude for the editorial decision to publish our manuscript in Life Science Alliance.

We have diligently addressed the constructive comments to conduct the final revisions to ensure compliance with LSA's formatting guidelines. Specifically:

1. Reviewer 2's minor comments:

- a. "In Fig 1, it's not clear what output of NetMHCpan is compared to, BA or EL?"

And similarly, what output of NeoMUST - presentation or affinity?"

We compared NeoMUST neoantigen presentation (NP) model with

NetMHCpan EL model in Fig 2. NetMHCpan EL and NeoMUST NP models

have been explicitly labeled in Fig 2 and other relevant figures to eliminate any ambiguity.

- b. "The figure legends should clearly indicate what the different data points in the scatter plots and boxplots represent (samples from 76 individuals?)"

The figure legends have been modified to clearly indicate the representation of different data points in the scatter plots and boxplots, specifying that they correspond to samples from 76 individuals.

- c. "Older methods like NetMHCpan-4.0 were not trained on as much presentation data, so why not just use the newer versions?"

Regarding the use of older versions like NetMHCpan-4.0, we have explained

in detail at page 7 the necessity to utilize NetMHCpan4.0 due to significant data leakage issues observed in the newer version, NetMHCpan4.1, for our primary benchmark test sets, TeSet-1 and TeSet-2. Additionally, we have constructed reduced datasets (TeSet-1-Filtered and TeSet-2-Filtered) for comparing the performance of NetMHCpan4.1, shown in the second paragraph on page 18 and Figure S3 and S6.

- d. “Instead of PPV at 1%, which is a particular snapshot of a precision-recall curve, the authors should use AUPRC as the performance metric.”

We have replaced PPV at 1% with AUC-PR in the main figures and texts.

Recognizing the importance of practical usage for top positive predictions, we have retained PPV at 1% in the supplementary figure for additional comparison.

January 22, 2024

RE: Life Science Alliance Manuscript #LSA-2023-02255-TRR

Dr. Hui Yao
Fresh Wind Biotechnologies USA Inc.
4502 Riverstone Blvd, STE1104
Missouri City, Texas 77459

Dear Dr. Yao,

Thank you for submitting your Research Article entitled "NeoMUST: an Accurate and Efficient Multi-Task Learning Model for Neoantigen Presentation". It is a pleasure to let you know that your manuscript is now accepted for publication in Life Science Alliance. Congratulations on this interesting work.

DISTRIBUTION OF MATERIALS:

Again, congratulations on a very nice paper. I hope you found the review process to be constructive and are pleased with how the manuscript was handled editorially. We look forward to future exciting submissions from your lab.

Sincerely,
